# Evaluation of Pseudo-Random Number Generation on GPU Cards

**Tair Askar** [1,2,*] , **Bekdaulet Shukirgaliyev** [2,3,4] , **Martin Lukac** [5] and **Ernazar Abdikamalov** [2,6]

1   School of Engineering and Digital Sciences, Nazarbayev University, Nur-Sultan 010000, Kazakhstan
2   Energetic Cosmos Laboratory, Nazarbayev University, Nur-Sultan 010000, Kazakhstan;
    bekdaulet.shukirgaliyev@nu.edu.kz (B.S.); ernazar.abdikamalov@nu.edu.kz (E.A.)
3   Fesenkov Astrophysical Institute, Almaty 050020, Kazakhstan
4   Department of Solid State Physics and Nonlinear Physics, Faculty of Physics and Technology,
    Al-Farabi Kazakh National University, Almaty 050040, Kazakhstan
5   Department of Computer Science, Nazarbayev University, Nur-Sultan 010000, Kazakhstan;
    martin.lukac@nu.edu.kz
6   Department of Physics, Nazarbayev University, Nur-Sultan 010000, Kazakhstan
*   Correspondence: tair.askar@nu.edu.kz

**Abstract:** Monte Carlo methods rely on sequences of random numbers to obtain solutions to many problems in science and engineering. In this work, we evaluate the performance of different pseudo-random number generators (PRNGs) of the Curand library on a number of modern Nvidia GPU cards. As a numerical test, we generate pseudo-random number (PRN) sequences and obtain non-uniform distributions using the acceptance-rejection method. We consider GPU, CPU, and hybrid CPU/GPU implementations. For the GPU, we additionally consider two different implementations using the host and device application programming interfaces (API). We study how the performance depends on implementation parameters, including the number of threads per block and the number of blocks per streaming multiprocessor. To achieve the fastest performance, one has to minimize the time consumed by PRNG seed setup and state update. The duration of seed setup time increases with the number of threads, while PRNG state update decreases. Hence, the fastest performance is achieved by the optimal balance of these opposing effects.

**Keywords:** GPU; PRNG; CUDA; Curand

## 1. Introduction

The rapid development of GPU cards in recent decades led to widespread adoption in many areas of scientific computing [1–6]. If efficiently utilized, the many-core architecture of the GPUs enables high-performance parallel computations. A class of widely used methods that can benefit from parallelization is the Monte Carlo (MC) method. This method relies on sequences of pseudo-random numbers (PRNs) to perform computations [7–9]. Examples of MC applications range from the evaluation of integrals to particle transport simulations [10–15]. One of the main components of MC calculations is the generation of PRN sequences with a given probability distribution. The most common PRNGs produce sequences with uniform distributions. Non-uniform distributions can be obtained using algorithms such as acceptance-rejection (AR), inverse-transform, and other methods [7].

There have been many studies of parallel PRN generation on multi-core CPUs [16–21], and GPUs [22–52]. The idea is to have different instances of PRNGs on different threads, each of which produce unique sequences of random numbers. This can be achieved in two fundamentally different ways: splitting and parameterization [21,22,32]. The splitting method divides the entire sequence of PRNs that a serial process would produce into sub-sequences and assigns them to different processors/threads. In the parameterization method, different processors/threads can run the same (or the same type of) PRNG with

distinct algorithm parameters so that they produce different samples. To maximize performance, the PRNG size should be small enough to fit into local memory. Reference [44] studied a GPU implementation of the Park-Miller PRNG using the CUDA platform, finding a speed-up by a factor of at least $44\times$ compared to the CPU implementation. Ref. [48] implemented PRNG on GPU for an MC particles transport code and observed up to $8.1\times$ speed-up compared to implementation on CPU with 4 to 6 cores. Furthermore, [53] studied the speed and quality of the proposed xorgenGP PRNG, finding a similar speed to the MTGP32 generator. Reference [54] proposed a GPU-based parallel PRNG on CUDA Fortran implemented using the global and shared memory modes. These are used to store a pre-generated table of seeds. Each thread runs PRNGs with unique seeds and generates independent parallel sequences of PRNs. They find a speedup of $150\times$ and $470\times$ over the single CPU core implementation for global and shared memory modes, respectively. However, compared to the Curand XORWOW generator, the proposed PRNG in shared memory mode was slower by 13%. In [55], authors present optimization of SHR3 generator to solve differential equations, which outperformed the Curand XORWOW generator by 79% and 38% for uniform and normal distributions, respectively. Reference [56] compared the performance of PRNGs from the VecRNG package (VecRNG is a part of VecMath, a collection of vectorized algorithms for high-energy physics applications based on the VecCore library [56,57].) with the Curand library PRNGs to generate $10^7$ double precision PRNs. They find that the Curand library implementation of the PHILOX4_32_10 generator is five times faster than the VecRNG implementation, while the MRG32k3a generators in both libraries show a similar performance.

The main aim of this work is to assess the feasibility of using GPU computing for MC calculations. We focus on the most fundamental component of the MC methods: generation of sequences of PRNs with uniform and non-uniform distributions, which often represents a significant fraction of the overall computational cost of the method. We extend the previous works by considering a wider range of performance measures, including the data transfer time between CPU and GPU as well as execution of API function calls (e.g., memory allocation, device synchronization, etc.), all of which consume resources and affect the overall performance [58]. We use a selection of modern Nvidia cards and consider five different PRNGs from the Curand library: XORWOW, PHILOX4_32_10, MTGP32, MT19937, and MRG32k3a. These generators produce PRNs with uniform distribution. Using the AR method, we produce three different non-uniform distributions: the Rayleigh, Beta, and Gamma distributions [59–61]. These numerical experiments serve as idealized test calculations that model the most fundamental component of MC methods.

We study how the performance depends on implementation parameters. To that end, we consider implementations on GPU and CPU as well as a hybrid GPU/CPU implementation, in which the uniform PRNs are generated on the GPU and transformed into non-uniform distribution on a CPU core using the AR method. For the GPU implementation, we additionally consider two different implementations using the host and device application programming interfaces (API). We analyze the dependence of performance on the number of threads per block and the number of blocks per streaming multiprocessor. Finally, we provide a comparison of implementations on different GPUs.

This work is organized as follows. In Section 2, we describe our methodology and computational setup. In Section 3, we present our results and in Section 4, we provide our summary and conclusions.

## 2. Methodology

### 2.1. GPU Architecture

Nvidia GPU architecture consists of a set of streaming multiprocessors (SMs), which contain different computing units such as CUDA cores, special function units, loading/storing units, and schedulers. Additionally, modern high-end GPU cards contain tensor and ray tracing cores designed for the acceleration of matrix multiplications and for graphical rendering of shadows and light respectively.

Nvidia GPU cards can be programmed by using the CUDA computing platform [62,63]. GPUs can execute instructions on multiple threads in parallel. The parallel part is executed as *kernels*. Each SM can accommodate a fixed maximum amount of threads (e.g., 1536 for RTX3090) grouped in thread blocks. Thread blocks are allocated by GPU schedulers to SMs once a kernel is started. SM divides blocks into warps. A warp is a basic processing unit that consists of 32 threads. SM issues instructions to be executed by threads in a warp simultaneously. Multiple warps can be processed in an SM at the same time.

To efficiently utilize a parallel structure of GPU, a complex memory hierarchy located on different access levels is available. The global memory is the main and largest memory of GPU cards, reaching 32 GB on modern GPUs. It is accessible by all threads, but it has a high access latency of up to several hundred clock cycles. The shared memory is smaller and has lower latency and higher bandwidth when compared to the global memory. It is accessed by threads within the same block. The size can be managed by software and can reach 96 KB on modern GPU cards. The registers are the fastest type of memory. In modern GPU cards, each SM has 256 KB of register memory. This memory is allocated among threads, so each thread has private registers. In addition, there are constant and texture memory spaces that are read-only and are declared from the host before running the GPU kernel. Constant memory can be used to keep constants and input values, while texture memory can be used to cache 2D and 3D data located in global memory. Constant memory size is limited to 64 KB, whereas texture memory is defined by array dimensions. Both of them can be accessed by all threads. More detailed information about memory structure can be found in, e.g., [63,64].

### 2.2. PRNG Parameters

The main parameters of the five different PRNGs that we consider in our work are summarized in Table 1. The MTGP32 generator uses the parameterization method, while XORWOW, MT19937, and MRG32k3a use the splitting method. PHILOX4_32_10 in the host API implementation uses the parameterization method, whereas in the device API implementation, it uses the splitting method [65]. The period of a PRNG tells us how many PRNs we can generate until it repeats itself. The length of the sub-sequence shows us how many PRNs can generate each thread/stream without overlapping with other threads/streams. More information about these PRNGs can be found in [65].

**Table 1.** Summary of the key parameters of the five Curand PRNGs considered in this work.

|  | **XORWOW** | **MTGP32** | **MRG32K3A** | **PHILOX4_32_10** | **MT19937** |
|---|---|---|---|---|---|
| Algorithm | Linear feedback shift registers [66] | Twisted generalized feedback shift register generator [67] | Combined Multiple Recursive [68] | Counter-Based Random Number Generation [69] | Twisted generalized feedback shift register generator [70] |
| Period | $2^{192} - 1$ | $2^{11,214}$ | $2^{191}$ | $2^{128}$ | $2^{19,937} - 1$ |
| Sub-sequence length | $2^{67}$ | – | $2^{67}$ | $2^{64}$ | $2^{1000}$ |
| State size | 48 bytes | 4120 bytes | 48 bytes | 64 bytes | 2500 bytes |
| Parallelization method | Sequence splitting | Parameterization | Sequence splitting | Sequence splitting, parameterization | **Sequence splitting** |

### 2.3. Experimental Setting

We perform calculations on a single AMD Ryzen Threadripper 3990X CPU core as well as on Nvidia GTX1080, GTX1080Ti, RTX3080, and RTX3090 GPU cards (see specifications of GPU cards in Table 2). We compare the Curand library PRNGs [65] using the same application, identical parameter sets (seed, ordering, etc.), and with a similar configuration of blocks and threads. Additionally, using different combinations of threads per block

on various GPU cards, we find the range of threads per block that allows us to achieve the highest performance. We use programming language C/C++ and CUDA computing platform version 11.4 and Nvidia driver version 470.57.02. To measure the execution time of computations, we use metric functions such as `clock_t`, `cudaEvent_t`, `nvprof`, and `nsys` profiling tools. All calculations are made in single-precision floating point format.

**Table 2.** Specifications of the Nvidia GPU cards considered in this work.

| | GTX1080 | GTX1080Ti | RTX3080 | RTX3090 |
|---|---|---|---|---|
| SMs | 20 | 28 | 68 | 82 |
| CUDA cores | 2560 | 3584 | 4352 | 10,496 |
| Max clock rate | 1.73 GHz | 1.58 GHz | 1.8 GHz | 1.7 GHz |
| Global memory | 8 GB | 11 GB | 10 GB | 24 GB |
| Theoretical performance | 8.873 TFLOPS (FP32) | 11.34 TFLOPS (FP32) | 29.77 TFLOPS (FP32) | 35.58 TFLOPS (FP32) |
| Bandwidth | 320.3 GB/s | 484.4 GB/s | 760.3 GB/s | 936.2 GB/s |

To study the performance of PRNGs on GPUs, we consider four different implementations: (1) The GPU implementation (`G_imp`), where the uniform PRNs are generated and then transformed into non-uniform distribution (i.e., Rayleigh, Beta, and Gamma distributions [59–61]) using the AR method on the GPU only. (2) The CPU implementation (`C_imp`), where all calculations take place on a single core of the CPU. (3) The hybrid implementation (`H_imp`), in which GPU generates uniform PRNs, then transfers them to the RAM, where the AR method is applied to obtain non-uniform distribution on a single core of the CPU, sequentially. (4) The GPU implementation with memory copy from the device to the host (`G_imp_mcpy`), to account for the time spent on transferring data from the GPU to the CPU at the end of the calculation.

We consider two different ways to call the PRNG functions in the `G_imp` implementation: using the host API and device API. For the former, the library functions are called on the CPU, but the computations are performed on the GPU. The PRNs can then be stored on the device's global memory or can be copied to the CPU for subsequent use. The device API allows us to call the device functions to set up PRNG parameters, such as seed, state, sequence, etc., inside a GPU kernel. The generated PRNs can be used immediately in the GPU kernel without storing them in global memory. To minimize the data movement within the GPU memory, we implement all PRNG steps (such as seed setup, state update, etc.) in the device API using a single kernel. In Section 3.3, we present a detailed comparison of these two API implementations.

To obtain a sequence of PRNs with a probability distribution $f(x)$ with the AR method, one generates two equal-length sequences of PRNs $x$ and $y$ with uniform distributions and then selects (accepts) the values of $x$ that satisfy the condition $y < f(x)$ [7]. The Beta, Gamma, and Rayleigh distribution functions that we consider in this work have acceptance rates of 0.67, 0.36, and 0.41, respectively. Figure 1 shows these functions for $0 \leq x \leq 1$ using normalization $f(x) \leq 1$.

As a main measure of performance, we measure the time it takes to execute the computations. All the provided measurements are based on average over 100 iterations. In our default fiducial setup, we use the MRG32k3a PRNG, Beta distribution, Curand device API implementation on an RTX3090 GPU card, as summarized in Table 3. In the following section, we vary each of these parameters to investigate the dependence on them. We point out that our default parameters are not meant to represent the best or the optimal parameters. Instead, it serves as a baseline parameter set, with respect to which we compare the performance measure for other values of these parameters.

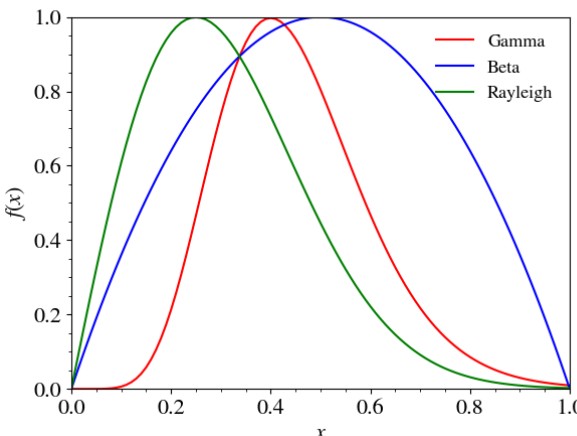

**Figure 1.** The probability density functions of the Beta, Gamma, and Rayleigh distributions (normalized to $f(x){\leq}1$). The areas under the curves of Beta, Gamma, and Rayleigh distributions are 0.67, 0.36, and 0.41, respectively.

**Table 3.** Default fiducial parameters of our experimental setup. The values of each of these parameters are varied to study the dependence of performance on these parameters.

| Parameter | Value |
|---|---|
| Implementation | `G_imp` |
| API | device API |
| GPU | RTX 3090 |
| PRNG | MRG32k3a |
| Distribution | Beta distribution |

## 3. Results

### 3.1. Comparison of GPU and CPU Implementations

Figure 2a shows the execution time as a function of the number of generated random candidate points ($N$) for different implementations. For small $N$ in the range of $N \lesssim 10^4$, the CPU implementation `C_imp` is faster `G_imp` and `H_imp` by up to two orders of magnitude. This difference in speed is caused by low occupancy of the GPU of ~3% (The GPU occupancy is defined as the number of active warps to the maximum number of warps that can run on the GPU). For small $N$, in contrast to the CPU cores, the GPU cores are not fully utilized. Since a single GPU core is slower than a single CPU core, partially-utilized GPU leads to slower execution compared to a single CPU. The `H_imp` implementation, in which the PRNs are generated on the GPU, but *AR-selected* in the CPU, performs similarly to the `G_imp` implementation. This is again caused by the low occupancy of the GPU, which becomes a bottleneck for this implementation.

The utilization of the GPU increases with $N$ and the GPU becomes faster than the CPU at $N \gtrsim 10^4$. At $N \gtrsim 10^8$, the GPU-based implementation is $\gtrsim 10^2$ times faster than the CPU one. The `H_imp` implementation performs faster than `C_imp` by a factor of seven, roughly, but performs slower than the `G_imp` implementation by a factor of $\simeq 46$.

The execution times of the `G_imp_mcpy` and the `G_imp` are similar to each other at $N \lesssim 10^6$, but the two diverge at larger $N$. This happens because the transfer of data from GPU to CPU becomes the bottleneck for the `G_imp_mcpy` implementation for $N \gtrsim 10^6$, whereas at lower loads the time spent on data transfer is negligible compared to the other components of the computation (e.g., seed setup, PRNG state update, and API function calls).

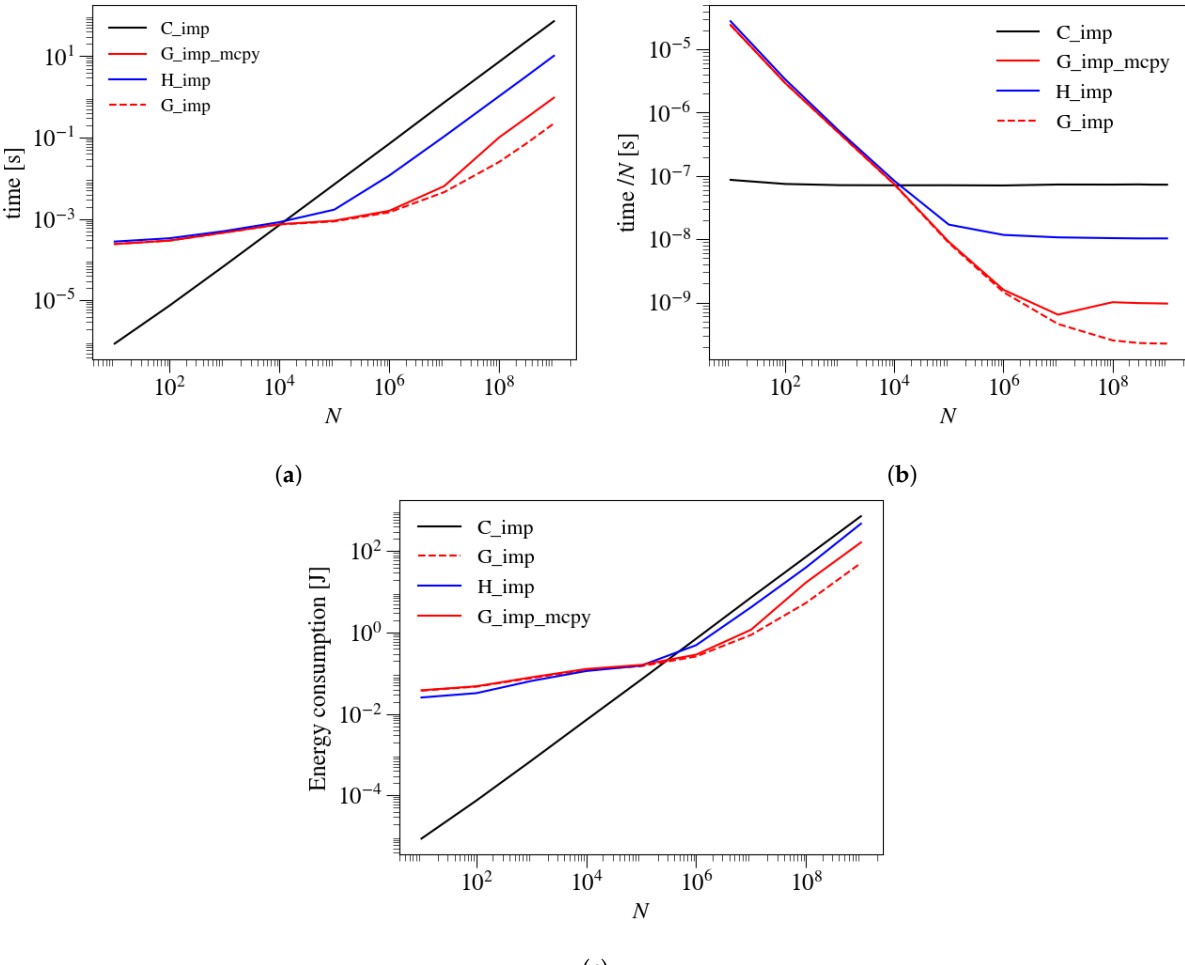

**Figure 2.** Execution time of computation (**a**), execution time per candidate point (**b**), and (**c**) an approximate energy consumption by different implementations as a function of ($N$) for Beta distribution. The solid black line represents the CPU implementation (`C_imp`); the dashed red line is for GPU implementation (`G_imp`), while the solid red line is for `G_imp_mcpy` implementation; the blue solid line is for hybrid implementation (`H_imp`).

These findings are reflected in the behavior of the execution time per candidate point as a function of $N$, shown in Figure 2b. The execution time per candidate point in the `C_imp` implementation is constant and independent of $N$, which shows that the CPU core is fully utilized for any $N$ (according to `htop` utility). In the `G_imp` implementation, due to the availability of many cores, the lowest execution time is reached at much larger $N$ of $\sim 10^8$. For `G_imp_mcpy`, the execution time per candidate point reaches its minimum at $N \sim 10^7$ due to the overhead caused by the GPU-to-CPU data transfer. For `H_imp`, this execution time decreases with $N$ until $N \sim 10^5$. At $N \gtrsim 10^5$, it becomes independent of $N$. Since for this implementation the AR algorithm is performed on the CPU, and since the saturation is reached at much higher $N$ for the GPU implementation `G_imp`, the independence of $N$ suggests that the CPU becomes the bottleneck of the computation for $N \gtrsim 10^5$ for the `H_imp` implementation.

Figure 2c shows a rough estimate of energy usage by different implementations as a function of $N$. Power usage of the GPU card was monitored through the utility program `nvidia-smi`. For the CPU, since it is fully loaded for any value of $N$ according to the data from `htop` utility, we use the maximum wattage of 10 W for a single AMD Threadripper 3990X core. The produce of power and the execution time yields the energy usage for various $N$. From Figure 2, we can conclude that for $N \gtrsim 2 \times 10^5$ the `G_imp` is preferable in terms of speed and energy efficiency, while for smaller $N$, the GPU does not offer an advantage neither in terms of energy usage nor execution time.

### 3.2. Operation-Wise Load Analysis

The total cost of the generation of non-uniform PRNs consists of several components: the seed setup, PRNG state update, the AR algorithm, and API function calls. Figure 3a shows the execution time of these components for the `G_imp` implementation, and Figure 3b shows the corresponding percentile fraction with respect to the total computation time.

For $N \gtrsim 10^7$, the seed time consumes more than $\sim$30% of the overall time, reaching $\sim$80% at $10^4 \lesssim N \lesssim 10^5$. The PRNG state update takes less than $\sim$10% for $N \lesssim 10^7$. The fraction of the PRNG state update grows to $\sim$40% for $N \sim 10^9$. The AR algorithm takes less than $\sim$20% for $N \lesssim 10^5$ but then grows with $N$ gradually and reaches $\sim$50% for $N \sim 10^9$. The rest of the execution time goes to API function calls, the fraction of which is $\sim$40% for $N \lesssim 10^2$, but drops below $\sim$20% for $N \gtrsim 10^3$. However, as we will see below, these findings depend on the PRNG parameters, including the seed setup and the state size. The `G_imp_mcpy` implementation, shown in Figure 3c,d, exhibits qualitatively similar breakdown of the total cost for $N \lesssim 10^6$. However, the contribution of data transfer to the GPU becomes dominant for $N > 10^6$ (e.g., at $N \sim 10^8$, it reaches $\sim$80%). Hence, the relative contributions of the rest of the computations, including the AR algorithm, become smaller compared to those in the `G_imp` implementation.

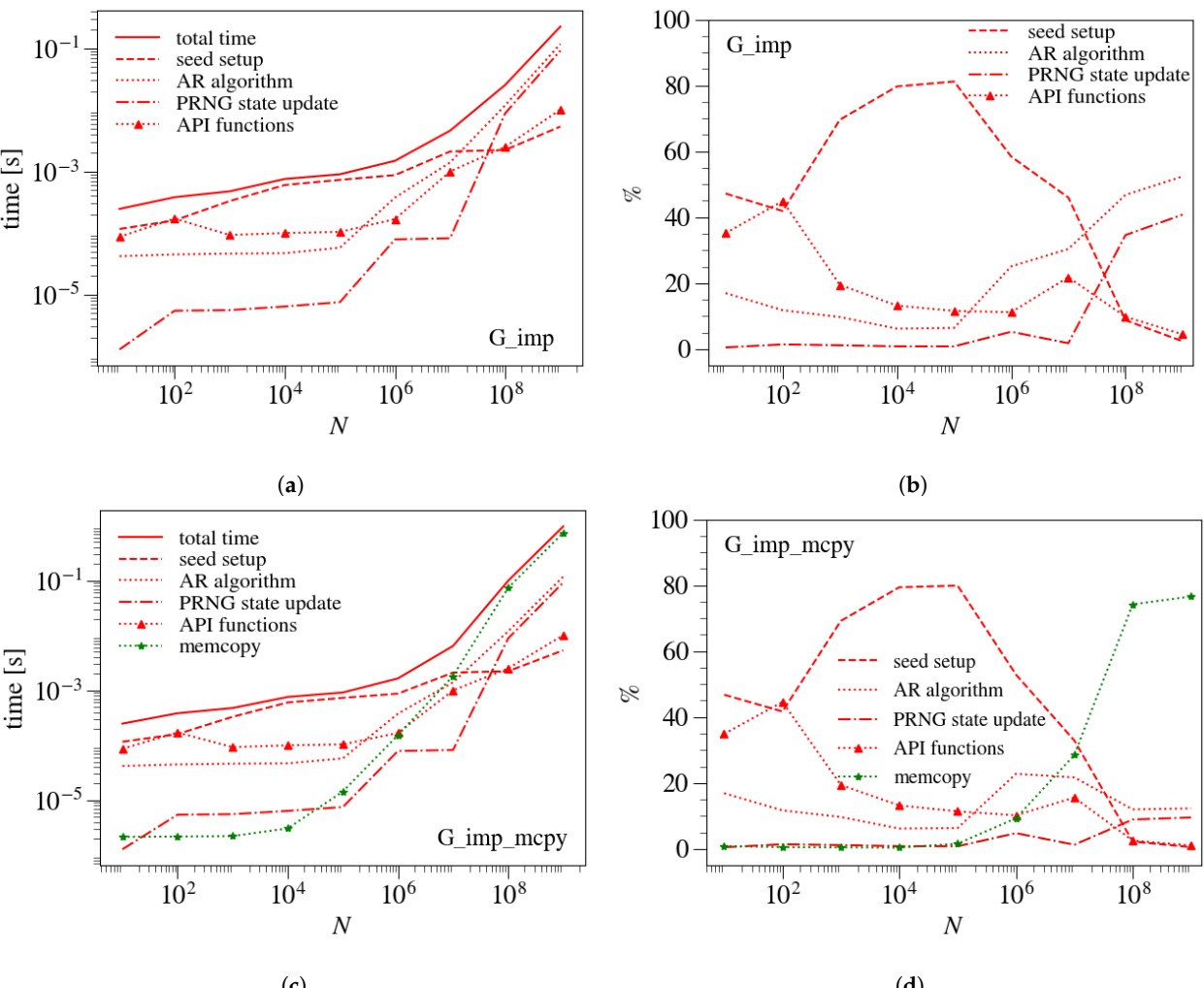

**Figure 3.** Execution time and percentage ratio of different computational parts as a function of $N$ for the `G_imp` (**a**,**b**) and `G_imp_mcpy` (**c**,**d**) implementations. The solid line represents the total computation time, the dashed line is PRNG seed setup time, the dotted line is AR algorithm time; the dashed-dotted line is the PRNG state update, and the dotted line with triangle marker is for the API functions. The green dotted line with star marker shows the memory copy time from device to host.

### 3.3. Device API and Host API Implementations

As mentioned above, we consider two different ways to call the PRNG functions in the `G_imp` implementation: using the host API and device API. Figure 4a shows the execution time for the Beta distribution in the host and device API implementations for the MRG32k3a PRNG. The device API is faster than the host API implementation by about an order of magnitude for $N \lesssim 10^6$. This is attributed to the PRNG seed setup time (shown in dashed blue line), which is a bottleneck for host API implementation for $N \lesssim 10^6$. The host API implementation of the MRG32k3a generator by default initializes 32,768 threads for any number of generated PRNs. Since each thread has its own state, the host API implementation has the same seed setup time of $\sim$6.1 ms for any $N$. It is possible to speed up the PRNG seed setup time by varying the ordering parameter, i.e., the parameter that determines how the PRNs are ordered in device memory [65]. For example, using `CURAND_ORDERING_PSEUDO_LEGACY` option for MRG32k3a can speed up the seed setup by a factor of $\sim$6 compared to the default parameter option. This is possible because fewer (4096) threads are initialized for this option. In the device API, for any $N$, we can choose a suitable number of initialized threads. As a result, the PRNG seed setup time in the device API implementation does not stay the same for any $N$. Instead, it increases with $N$ from $\sim$0.116 ms at $N = 10$ to $\sim$5.36 ms at $N = 10^9$ (red dashed line in Figure 4a). Acceleration of the seed setup by changing the ordering option has also been discussed in [15,62].

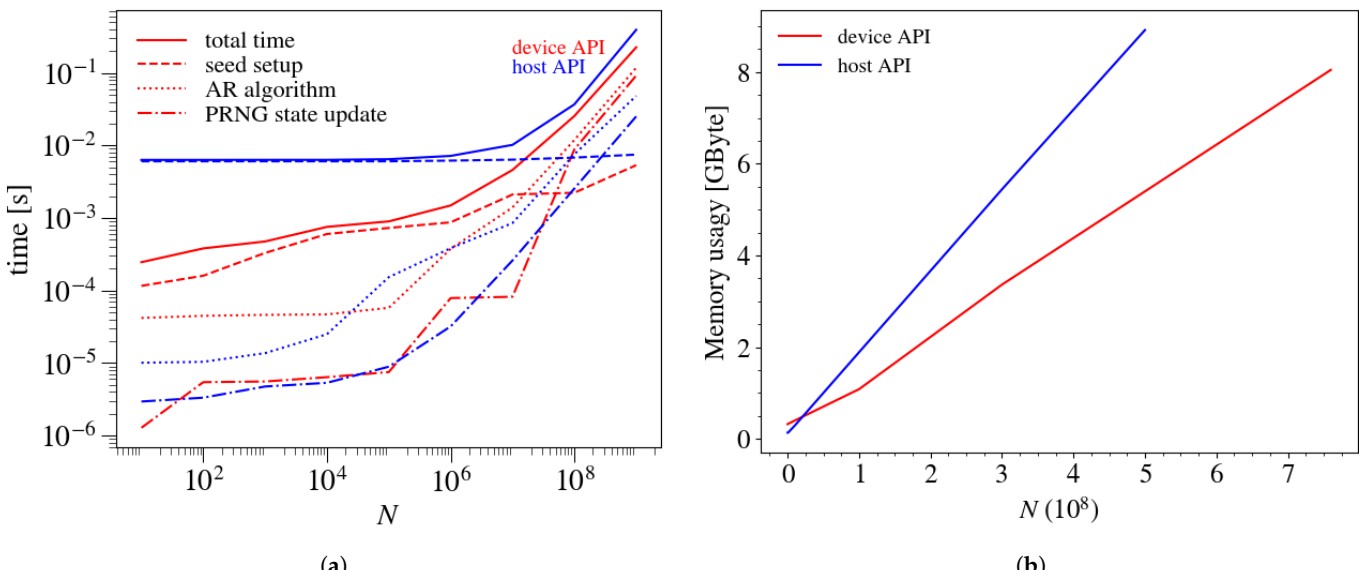

**Figure 4.** The execution time (**a**) and the GPU global memory usage (**b**) as a function of $N$ using host and device API implementations. In (**a**) the solid lines represent the total execution time, the dashed lines show seed setup time, and the dashed-dotted lines show the PRNG state update time; the dotted lines display time spent to apply the AR algorithm.

The execution time of the PRNG state update and AR algorithm (shown with dash-dotted and dotted lines, respectively) increases with $N$ for both implementations. Due to this reason, the relative importance of the seed setup decreases with increasing $N$. As a result, the difference in the execution time between the device API and host API implementations decreases to eventually becoming a factor of $\sim$1.45 at $N \sim 10^8$.

Figure 4b shows the global memory usage for the host API and device API implementations. Since the host API implementation requires storing the generated PRNs in global memory, it uses more memory than the device API implementation during the course of computation. The device API implementation allows us to use generated PRNs immediately without storing them in global memory. Thus, if PRNs need to be reused several times or copied to the CPU, then the host API implementation is a better choice.

Otherwise, device API should be used in order to save memory space and to avoid extra loading and storing operations to global memory.

### 3.4. Comparison of Different PRNGs

Figure 5 shows the execution time for different PRNGs as a function of $N$ for the host (Figure 5a) and device (Figure 5b) API implementations. Since in this section we compare different PRNGs, we consider only uniform distribution sequences in order to exclude the impact of the AR calculations. Note that since the MT19937 implementation does not support device API [65], it is omitted from Figure 5b. As discussed above for the MRG32k3a PRNG (cf. Section 3.3), the longer seed setup time in the host API implementation leads to longer execution times, especially at lower $N$. We find a similar difference between the two implementations for the rest of the PRNGs considered here.

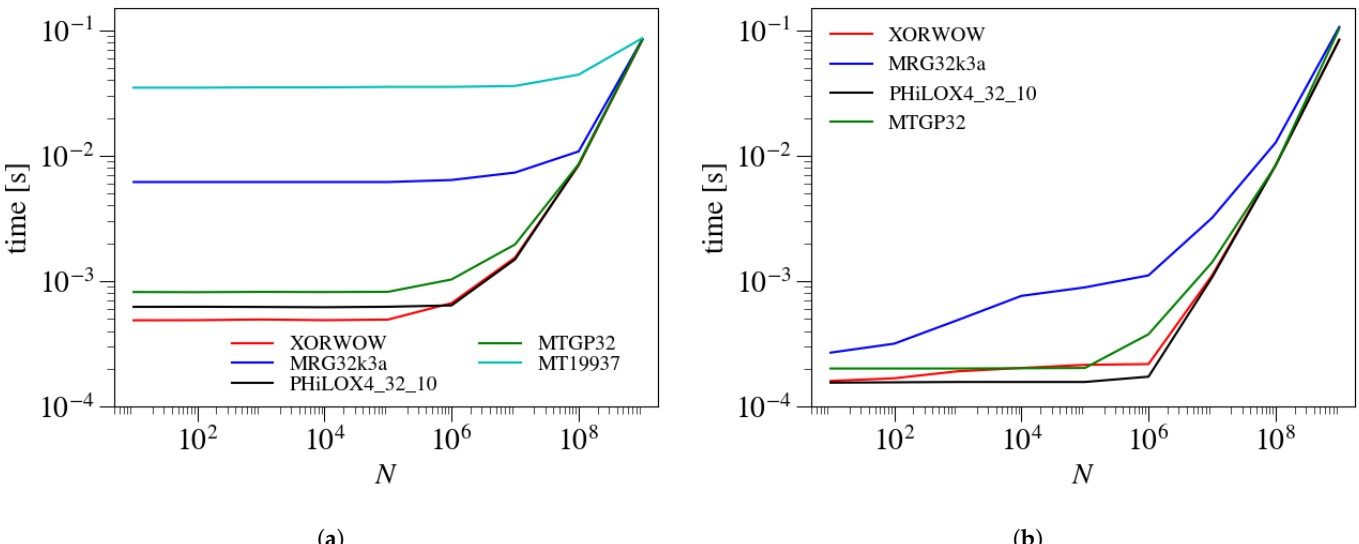

(**a**)                                                                                    (**b**)

**Figure 5.** Execution time for different PRNGs as a function of $N$ using host API (**a**) and device API (**b**). Note that since the MT19937 implementation does not support device API [65], it is omitted from (**b**).

In both implementations, the XORWOW and PHILOX4_32_10 are the fastest generators followed by MTGP32. The XORWOW generator uses the XOR and shift bitwise operations which require one clock cycle for execution [71], while PHILOX4_32_10 is a non-state PRNG (i.e., it uses thread ID as its state) based on counter to generate PRNs, which enables faster speeds [69]. The slowest generators are MT19937 for the host API implementation (as mentioned earlier, this PRNG is not supported in the device API implementation), while the MRG32k3a generator is the slowest in the device API implementation. The big state size of MT19937 of 2.5 kB leads to low occupancy of SM cores because of the limited size of local memory per SM. The modulus operator of MRG32k3a generator has a latency of 22–29 clock cycles [63,71], which leads to longer execution times.

The difference in speed between different PRNGs is particularly strong at $N \lesssim 10^7$, but becomes small for $N \gtrsim 10^8$. This is because the API functions become the dominant contributor to the execution time of the fastest three PRNGs (XORWOW, PHILOX4_32_10, and MTGP32) (cf. Figure 6), which brings them closer in speed to the slowest two PRNGs (MT19937 and MRG32k3a). For MT19937, the seed setup is the most expensive component of the overall computation for $N \lesssim 10^7$. For PHILOX4_32_10, almost 100% of computational time is consumed by the API function calls in the device API implementation.

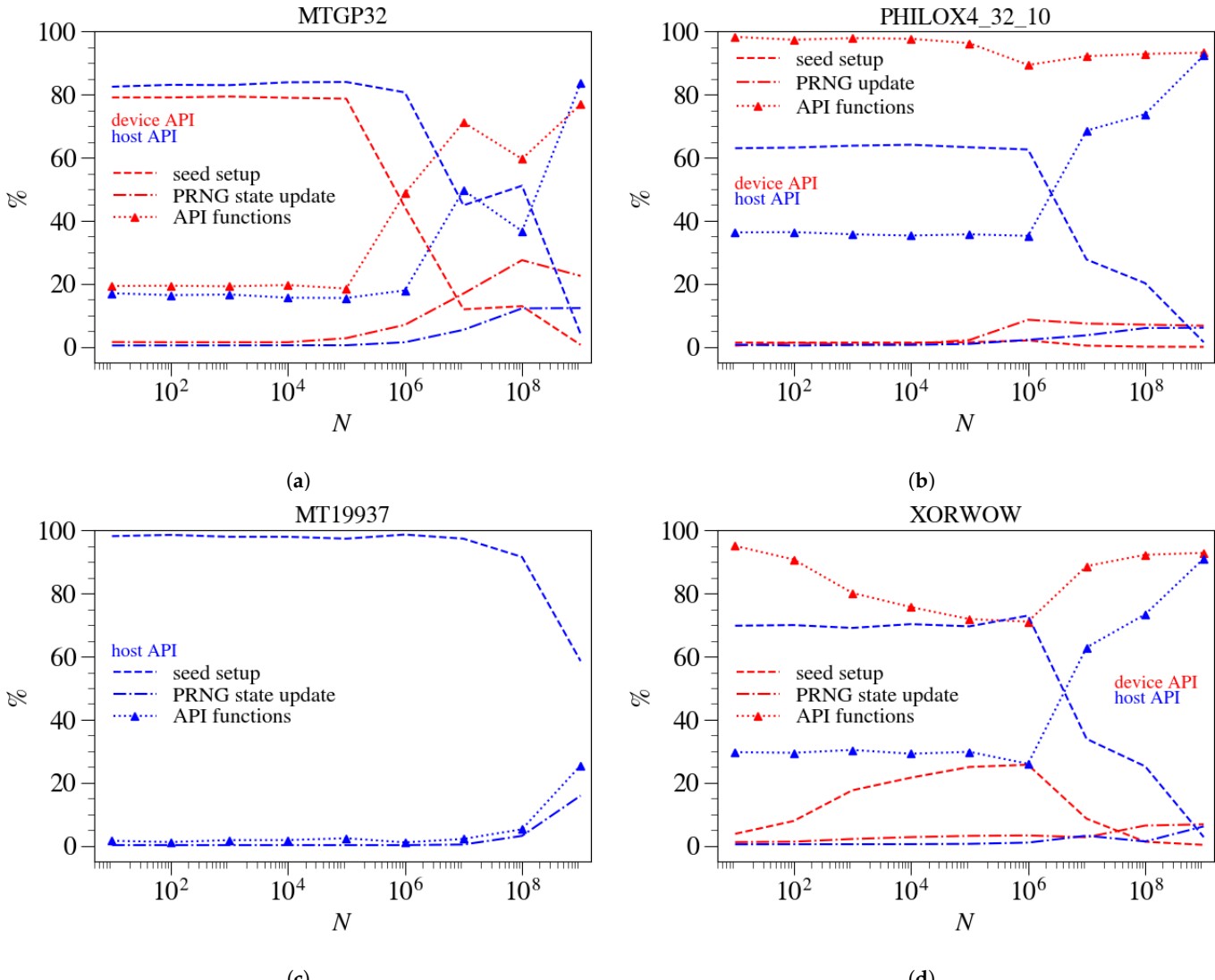

**Figure 6.** Percentage ratio of different computational parts as a function of $N$ for various PRNGs. (**a**) MTGP32 (**b**) PHILOX4_32_10 (**c**) MT19937 (only supported by host API) (**d**) XORWOW. The dashed lines represent the PRNG seed setup time; the dashed-dot lines are PRNG state update time, and the dotted lines with triangle markers are for API functions.

Figure 7 shows the normalized time as a function of GPU occupancy for the four PRNGs (note that since the MT19937 implementation does not support device API [65], it is omitted from this analysis). The time is normalized to its lowest value that can be reached for this PRNG and GPU for a given $N$. The PHiLOX4_32_10 generator reaches the fastest performance when the GPU occupancy is 100%. This is due to a combination of two reasons. First, this PRNG has a small state (cf. Table 1), so we can fit more threads into the GPU memory. Second, this PRNG has a small seed setup time (cf. Figure 6), so increasing thread number, which leads to a longer seed setup, will not negatively affect the overall performance. Instead, more threads lead to faster PRNG state update and AR calculations due to enhanced parallelism. For XORWOW PRNG, the fastest performance is reached at occupancy of ∼5% for $N = 10^7$ and $N = 10^5$, while for $N = 10^9$, the execution is not sensitive to occupancy.

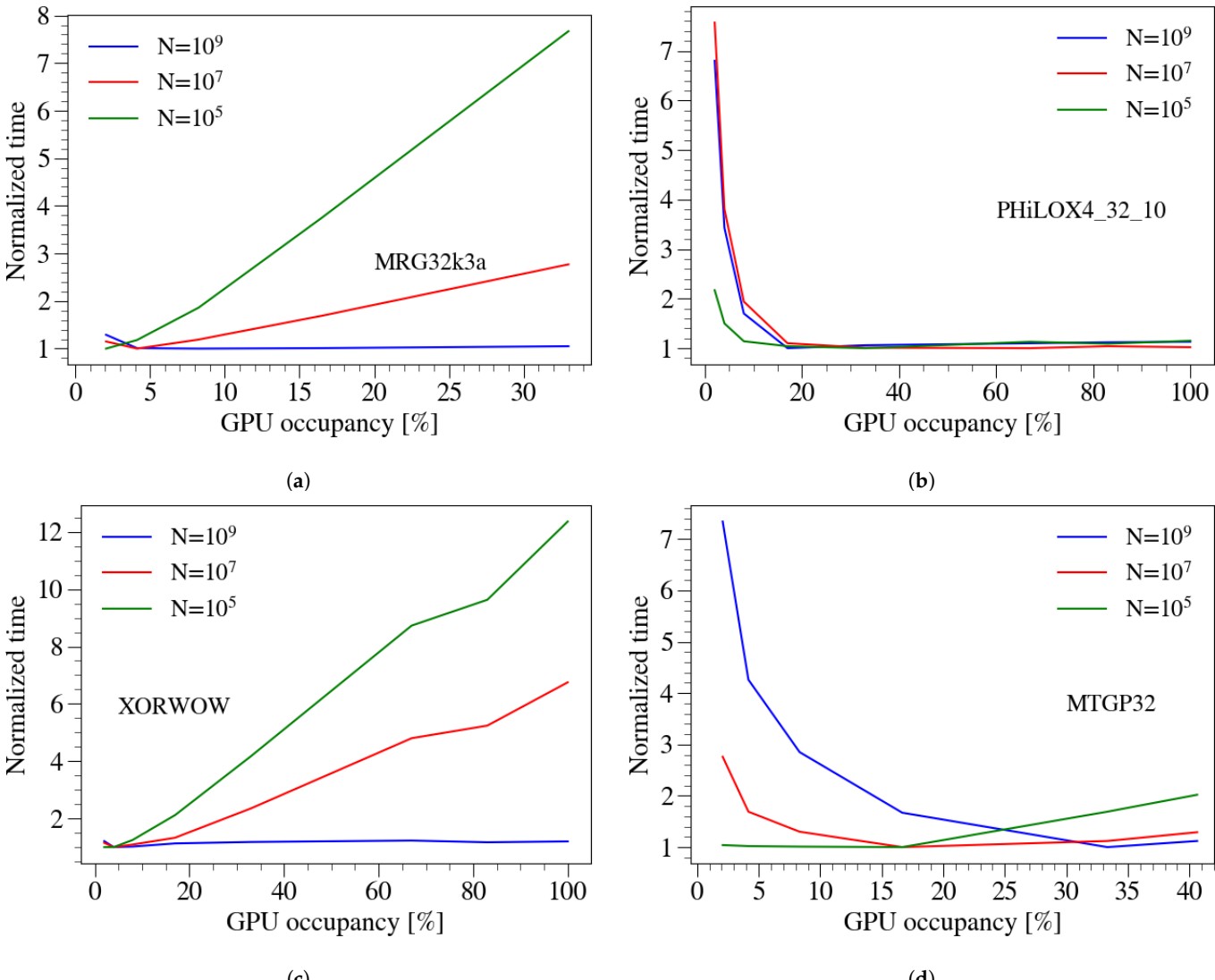

**Figure 7.** Normalized execution time versus the GPU occupancy for different PRNGs using device API implementation. (**a**) MRG32k3a; (**b**) PHILOX4_32_10; (**c**) XORWOW; (**d**) MTGP32. The time is normalized with respect to the fastest time that can be achieved for a given *N* for RTX3090 card and device API implementation. The various lines correspond to different numbers of generated PRN with the uniform distribution. The GPU occupancy is the ratio of active warps on a streaming multiprocessor to the maximum possible warps that can be run simultaneously.

The situation is dramatically different for the other two PRNGs. For example, the MTGP32 generator never reaches occupancy above 41% because of the limited number of blocks and threads that can be used to generate PRNs according to the implementation of this PRNG on GPU [65]. The MRG32k3a stays below 35% occupancy. This is caused by memory intensive modulus operator that is used for seed setup and state update for this generator [63,68]. The occupancy does not increase with increasing *N* beyond $N \simeq 10^9$, as demonstrated in Figure 8.

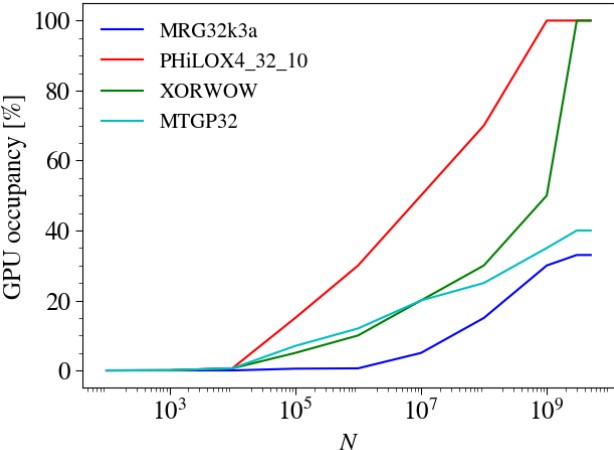

**Figure 8.** GPU occupancy for different PRNGs to generate various numbers *N* of PRNs implemented in device API.

### 3.5. Comparison of Different GPU Cards

Figure 9 shows the execution time as a function of *N* for the default fiducial parameters for different GPU cards. Since GPU cards are under-utilized for small $N \lesssim 10^4$, different cards yield similar results. As the occupancy of the card grows with increasing *N*, different cards yield varying execution times. The fastest card is RTX3090, followed by RXT3080, GTX1080Ti, and GTX1080. This correlates with the number of cores available in these cards. Moreover, the RTX cards support the newer Ampere architecture compared to the Pascal architecture of the latter two.

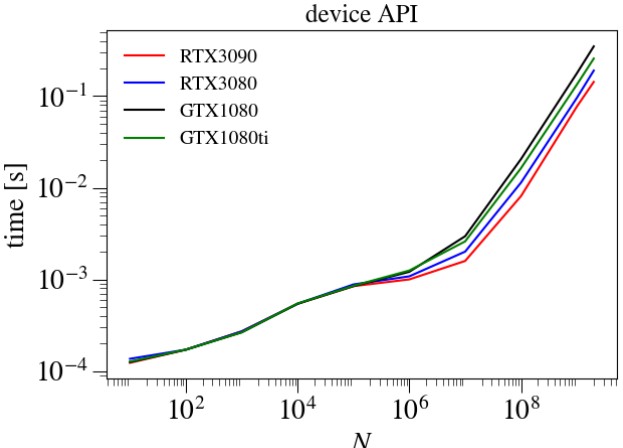

**Figure 9.** Execution time to generate PRNs as a function of *N* for the Beta distribution and MRG32k3a generator for different GPU cards.

### 3.6. Dependence on Distributions

Figure 10 shows the execution time for uniform and non-uniform distributions for the `G_imp` implementation as a function of the number of "accepted" PRNs ($N_a$). Since the uniform distribution does not require the application of the AR algorithm, it is the fastest to compute. For example, for $N_a = 10^6$, it is faster than the Gamma, Rayleigh, and Beta distributions by a factor of 2.78, 1.90, and 1.60, respectively. The different execution times of the non-uniform distributions are the direct result of the different acceptance rates of each distribution to construct non-uniform distributions (cf. Section 2.3). The execution times of different distributions diverge further with growing *N*. The reason for this behavior is that at smaller $N_a$, the contribution of state setup and API function calls to the overall execution time dominates over the contributions of the AR algorithm and the PRN state

update algorithm. At large $N_a$, the contributions of the latter two become more pronounced, leading to stronger differences. For example, at $N_a = 10^8$, the uniform distribution is faster than the Gamma, Rayleigh, and Beta distributions by factors of 10.65, 4.25, and 2.49, respectively.

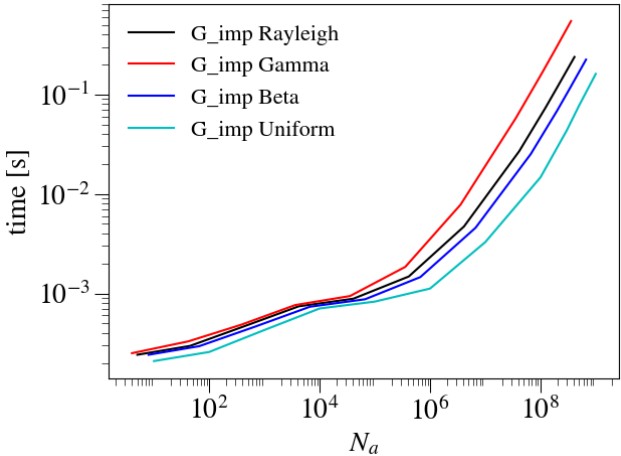

**Figure 10.** Execution time to obtain PRNs with different distributions using the device API implementation and the MRG32k3a generator.

## 4. Conclusions

In this work, we evaluated the performance of different parallel pseudo-random number generators (PRNGs) on Nvidia GPU cards GTX1080, GTX1080Ti, RTX3080, and RTX3090. We consider five different PRNGs from the Curand library (XORWOW, PHILOX4_32_10, MTGP32, MT19937, and MRG32k3a). We generate sequences of random numbers with uniform distribution and transform them into non-uniform distributions using the acceptance-rejection (AR) method. We consider different implementations of the PRNGs: single-core CPU implementation (using AMD Ryzen Threadripper 3990X CPU), GPU implementation with and without data transfer from the GPU to the CPU host, and a hybrid CPU/GPU implementation, where PRNG are produced on a GPU and transformed to non-uniform sequence on a CPU using the AR method. For the GPU implementation, we additionally consider implementations via the host and device APIs. We consider different configurations of these parameters and study how the performance is affected with respect to our default configuration (cf. Table 3).

We find that when the length of the pseudo-random number (PRN) sequence is less than $\sim 10^4$, a single CPU core is faster than the GPU. In this regime, unlike the CPU core, the GPU cores are not fully loaded. Since the single CPU core is much faster than the GPU core, sparsely loaded GPU yields a slower result than a fully-loaded CPU. For PRNs more than $\sim 10^4$, the GPU cores are loaded more evenly, leading to faster performance. For $\gtrsim 10^6$ PRNs, the GPU is faster than the CPU by two orders of magnitude.

To achieve the fastest performance on a GPU, one has to minimize the duration of the PRNG seed setup and state update. The former increases with the number of parallel threads, while the latter decreases. The fastest performance is achieved by the balance of these opposing effects. The right number of parallel threads depends on the number of PRNs that need to be generated. If the latter far exceeds the number of cores on a GPU (e.g., $\gtrsim 10^7$), it is desirable to have the largest possible number of threads that can fit GPU. For such configuration, the relative importance of seed setup is minor compared to the overall execution time. For a smaller sequence of PRN, the number of parallel threads that yield the fastest performance is a compromise between the seed setup and state update durations and it is different for different PRNGs considered in this work. The findings will help us to choose the optimal performance parameters for our future scientific computing

applications, including Monte Carlo radiative transfer simulations that we plan to perform in our future work.

**Author Contributions:** Writing—original draft preparation, T.A.; writing—review and editing, B.S., M.L. and E.A. All authors have read and agreed to the published version of the manuscript.

**Funding:** This research was partially funded by the Ministry of Education and Science of the Republic of Kazakhstan.

**Institutional Review Board Statement:** Not applicable.

**Informed Consent Statement:** Not applicable.

**Acknowledgments:** This research has been funded by the Science Committee of the Ministry of Education and Science of the Republic of Kazakhstan (AP08856149, BR10965141).

**Conflicts of Interest:** The authors declare no conflict of interest. The funders had no role in the design of the study; in the collection, analyses, or interpretation of data; in the writing of the manuscript, or in the decision to publish the results.

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
