# Peer review of "Evaluation of Pseudo-Random Number Generation on GPU Cards"

_computation, doi:10.3390/computation9120142_

Round 1

Reviewer 1 Report

This paper evaluates the performance of different pseudo-random number generators (PRNGs) of the Curand library. Experiments consist of generating PRNGs sequences with non-uniform distributions, and were performed on CPUs, GPUs, as well as on hybrid (CPU & GPU) platform setups. Authors conclude that the highest performance is achieved by optimally balancing the times consumed by PRNG seed setup and state update phases.

The article is well-organized, descriptions and figures are very illustrative, while experiments are sound and clear. However, I think the article requires clarifications (see below).

Questions:

  1.  In Table 1: MTGP32 and MT19937 are parallelized via parametrization. If I understand correctly, since they do not use splitting, then the sub-sequence length in both cases should be "not applicable", shouldn"t it ? If so, then why MT19937 has a sub-sequence length of 2^1000 ?
  2.  Line 106: Regarding the "C_imp": why not comparing the performance of a GPU-implementation vs. a multi-core CPU implementation (i.e., 64-core ThreadRipper 3990X CPU instead of using a single core)?
  3. Line 149: What is the definition of cost ? if it is "execution time per candidate point", then fix parenthesis position: "cost (i.e., exec... ...e point)"
  4. Line 150: "..., which shows that the CPU core is fully utilized for any N." Did you profile the execution on CPUs? It appers that this is inferred rather than meaured. Either way, please indicate that.
  5. Regarding programing details: are the PRNG steps (seed setup, PRNG state update, etc) implemented as different GPU kernels ? Maybe authors can give us more details on the GPU program structure ?
  6. Line 158: "GPU consumes about 35 times more energy than single CPU core" -> that is not necessarily true. Reason: "Energy = Power x Time". Please calculate the energy and update your statement on the energy comparison.
  7. Line 167: According to the plot, it appears that text description is incorrect. Please double check it.
  8. Line 171: Any explanation on why for N > 10^6, the time percentage of the AR algorithm is different in these two cases (G_imp vs. G_imp_mcpy) ?
  9. Line 181: Will the seed setup time be shorter if multiple CPU cores were leveraged ?
  10. Line 192-193: If I understand correctly, at N ~ 10^8, there is a small performance advantage (1.45x) of GPU over single-core CPU. So, would it make sense to generate random numbers on GPUs -- instead on CPU -- only for N > 10^8 ?
  11. Line 212-214: This requires clarification, because wasn't the MT19937 only supported on the host (as stated in lines 203-204)?

Minor fixes:

  • Missing space in first footnote: "library<SPACE>[35,36]"
  • Table 2: besides the provided specs, it would be useful to also indicate the max. theoretical performance (TFLOPS) and max. bandwith (GB/s)
  • Line 160: if appropriate, I would suggest to adapt it accordingly "...advantage neither in terms of power usage/energy nor execution time."

Author Response

Good day, dear reviewer.

Please find attached the pdf file with responses.

Thank you and have a nice day.

Reviewer 2 Report

The main aim of this work is to assess the feasibility of using GPU computing for MC calculations. We focus on the most fundamental component of the MC methods: generation of sequences of PRNs with uniform and non-uniform distributions, which often represents a significant fraction of the overall computational cost of the method. We extend the previous works by considering a wider range of performance measures, including the data transfer time between CPU and GPU as well as execution of API function calls (e.g., memory allocation, device synchronization, etc.), all of which consume resources and affect the overall performance [37]. We use a selection of modern Nvidia cards and consider five different PRNGs from the Curand library: XORWOW, PHILOX4_32_10, MTGP32, MT19937, and MRG32k3a. These generators produce PRNs with uniform distribution. Using the AR method, we produce three different non-uniform distributions: the Rayleigh, Beta, and Gamma distributions. These numerical experiments serve as idealized test calculations that model the most fundamental component of MC methods. We study how the performance depends on implementation parameters. To that end, we consider implementations on GPU and CPU as well as a hybrid GPU/CPU implementation, in which the uniform PRNs are generated on the GPU, and transformed into non-uniform distribution on a CPU core using the AR method. For the GPU implementation, we additionally consider two different implementations using the host and device application programming interfaces (API). We analyze the dependence of performance on the number of threads per block and the number of blocks per streaming multiprocessor. Finally, we provide a comparison of implementations on different GPUs.

I like the article and consider it partially original.

I consider the presented article to be original and it contains a considerable amount of used literature.

In the present article would also be appropriate to indicate the planned continuation of research in the area.

Author Response

(The authors gave the same response as above.)

Reviewer 3 Report

This topic is interesting because it concerns an issue very close to security systems, a very hot topic. The manuscript is well structured, but not completely exhaustive. In this manuscript, there is not a section describing the current state of the art, approaches and techniques used and already well known and to follow a section of comparison and discussion between the results obtained in this work compared to those already present in the literature. In this version it is difficult to understand what the real innovative part is, the scientifically relevant and interesting part. This aspect needs to be highlighted more. This reviewer thinks that the current version is fine, there are no changes to be made but parts to add and comment comprehensively.

Author Response

(The authors gave the same response as above.)

Round 2

Reviewer 3 Report

All my suggestions have been addressed. No further changes are necessary.